# The Safety and Efficacy of Glibenclamide in Managing Cerebral Edema After Aneurysmal Subarachnoid Hemorrhage: A Systematic Review and Meta-Analysis

**DOI:** 10.3390/brainsci15070677

**Published:** 2025-06-24

**Authors:** Majd M. AlBarakat, Rana B. Altawalbeh, Khaled Mohamed Hamam, Ahmed A. Lashin, Ahmed Wadaa-Allah, Ayah J. Alkrarha, Mohamed Abuelazm, James Robert Brašić

**Affiliations:** 1Faculty of Medicine, Jordan University of Science and Technology, Irbid 22110, Jordan; mmalbarakat20@med.just.edu.jo (M.M.A.); rbaltawalbeh20@med.just.edu.jo (R.B.A.); ayahalkrarha@gmail.com (A.J.A.); 2College of Health and Human Sciences, North Dakota State University, Fargo, ND 58102, USA; khaled.mohamed@ndsu.edu; 3Faculty of Medicine, Benha University, Benha 13511, Egypt; ahmed180511@fmed.bu.edu.eg; 4Department of Biochemistry, Faculty of Science, Ain Shams University, Abbassia, Cairo 1181, Egypt; a.m.wadaa-allah.p@sci.asu.edu.eg; 5Faculty of Medicine, Tanta University, Tanta 31527, Egypt; dr.mabuelazm@gmail.com; 6Department of Psychiatry, New York City Health and Hospitals/Bellevue, New York, NY 10016, USA; 7Department of Psychiatry, New York University Grossman School of Medicine, New York University Langone Health, New York, NY 10016, USA; 8Section of High-Resolution Brain Positron Emission Tomography Imaging, Division of Nuclear Medicine and Molecular Imaging, The Russell H. Morgan Department of Radiology and Radiological Science, The Johns Hopkins University School of Medicine, Baltimore, MD 21287, USA

**Keywords:** calcium channel blocker, clinical trials, endothelium, functional independence, hypoglycemia, morbidity, mortality, relative risk, stroke, sulfonylurea receptor 1-transient receptor potential melastatin 4 channel blocker

## Abstract

Background/Objectives: We sought to determine if glibenclamide, a sulfonylurea receptor 1-transient receptor potential melastatin 4 (SUR1-TRPM4) channel blocker, reduces cerebral edema and improves neurological functioning in aneurysmal subarachnoid hemorrhage (aSAH). Methods: Following Preferred Reporting Items for Systematic reviews and Meta-Analyses (PRISMA) guidelines, a systematic search was conducted in PubMed, Cochrane Library, Web of Science, and SCOPUS for studies evaluating glibenclamide in aSAH patients. Primary outcomes included scores on the modified Rankin Scale (mRS) at discharge and the Subarachnoid Hemorrhage Early Brain Edema Score (SEBES) at ten days post-intervention. Secondary outcomes included adverse events, and safety and efficacy endpoints. Random-effects models were employed for meta-analyses. Results: Three studies utilizing oral glibenclamide (*n* = 245) met inclusion criteria. Oral glibenclamide demonstrated no significant improvements in mRS scores [MD −0.19 with 95% CI (−2.05, 1.66)] at discharge, [MD 0.06, (−0.60, 0.71)] at 3 months, and [MD 0.4, (−0.67, 0.87)] at 6 months; functional independence [risk ratio (RR) 1.05, (0.81, 1.36)]; independent ambulation [RR 1.07, (0.77, 1.48)]; mortality [RR 0.79, (0.42, 1.50)]; or delayed cerebral ischemia [RR 0.58, (0.31, 1.09]). Hypoglycemia risk was significantly higher in the glibenclamide group [RR 3.92, (1.14, 13.49)]. Conclusions: Oral glibenclamide offers a novel approach to addressing cerebral edema in aSAH but shows limited clinical efficacy in improving functional and neurological outcomes in subtherapeutic doses. Its safety profile is acceptable, though hypoglycemia risk necessitates careful monitoring. Further research is required to optimize dosing, timing of intervention, and patient selection to enhance therapeutic outcomes. By contrast, intravenous administration of therapeutic doses of glibenclamide offers a promising avenue for future studies in the management of aSAH by taking advantage of the favorable pharmacokinetics of this route of administration.

## 1. Introduction

Aneurysmal subarachnoid hemorrhage (aSAH), a severe neurological emergency that accounts for approximately 5% of all strokes, results from intracranial aneurysm rupture in 85% of cases [1,2,3]. Striking clinical and economic burdens result from aSAH because aSAH afflicts younger populations than other stroke subtypes [3,4]. Despite advances in aneurysm repair and the use of nimodipine, a selective calcium channel blocker, aSAH continues to carry high morbidity and mortality rates, with nearly 44% of patients succumbing to the condition and many survivors experiencing long-term cognitive dysfunction and poor neurological outcomes [4,5,6].

The neurological complications of aSAH result from the increased intracranial pressure (ICP) due to cerebral edema [5,6]. The management of global cerebral edema remains challenging, as decompressive craniectomy has shown limited long-term benefits, and nimodipine provides only modest improvements in clinical outcomes [7,8,9]. Therefore, there is a growing need for novel therapies targeting the molecular pathways involved in cerebral edema.

Glibenclamide (5-chloro-*N*-[2-[4-(cyclohexylcarbamoylsulfamoyl)phenyl]ethyl]-2-methoxybenzamide, C_23_H_28_ClN_3_O_5_S) [10,11] (Figure 1), a sulfonylurea receptor 1-transient receptor potential melastatin 4 (SUR1-TRPM4) channel blocker, has shown neuroprotective effects by reducing edema and maintaining the integrity of the endothelium and the blood–brain barrier [12,13] in preclinical models of brain injury [14,15]. The SUR1-TRPM4 channel, which is overexpressed following ischemic and hemorrhage insults, plays a critical role in cytotoxic and vasogenic edema by regulating sodium and water influx into brain tissues [16,17]. Blocking this channel has been shown to reduce secondary neuronal damage and improve outcomes in models of ischemic and hemorrhage strokes [14,15].

Despite promising preclinical results, glibenclamide has not been thoroughly evaluated in the context of aSAH. The 3- to 10-day window from the onset of aSAH to vasospasm and delayed cerebral ischemia (DCI) presents an opportunity for early pharmacological intervention [18]. Given the limited efficacy of current treatment strategies, we sought to conduct a systematic review and meta-analysis with the aim of synthesizing the available evidence regarding the safety and efficacy of oral glibenclamide in mitigating cerebral edema and improving clinical outcomes following aSAH. Establishing the therapeutic role of oral glibenclamide could offer new insights into managing ICP and early brain injury, potentially improving long-term outcomes in aSAH patients.

## 2. Materials and Methods

### 2.1. Protocol Registration

This systematic review and meta-analysis was conducted in accordance with the Preferred Reporting Items for Systematic Reviews and Meta-Analyses (PRISMA) guidelines and the Cochrane Handbook of Systematic Reviews and Meta-Analyses [19,20]. The study protocol was registered in the International Prospective Register of Systematic Reviews (PROSPERO) under the registration number CRD42024591036.

### 2.2. Data Sources and Search Strategy

A systematic search was conducted across four databases: PubMed, Cochrane Library, Web of Science (WOS), and SCOPUS. The search was carried out by two independent reviewers, M.M.A and M.A., without the application of search limits, to ensure the inclusion of all relevant studies published by 17 August 2024. Details of the search strategy for each database are provided in Table 1.

### 2.3. Eligibility Criteria

The review included studies that focused on patients with subarachnoid hemorrhage (SAH) as the target population. Glibenclamide was the intervention under investigation, with a placebo serving as the comparator. The primary outcomes assessed were the discharge modified Rankin Scale (mRS) score [21] and the Subarachnoid Hemorrhage Early Brain Edema Severity (SEBES) score [22], evaluated at 10 days post-intervention with a target score of 0–2. Secondary outcomes included the mRS score at discharge, at 3 months, and at 6 months, the modified Fisher Scale (mFS) [23] at 10 days after intervention, and the occurrence of adverse events. Any additional efficacy and safety data reported were also considered relevant. Both randomized controlled trials (RCTs) and observational studies were eligible for inclusion in the review. Studies were excluded if they involved animal models, lacked a control group, did not report relevant clinical outcomes, were not published in English, or were case reports, editorials, or conference abstracts

### 2.4. Study Selection

Three reviewers, A.A.L, A.J.A., and R.B.A., independently screened the titles and abstracts of the identified studies using Covidence [24] software after duplicates were removed. The same reviewers then conducted full-text screening to assess eligibility based on the predefined criteria. Any disagreements during the screening process were resolved through discussion among the reviewers.

### 2.5. Data Extraction

Data extraction was performed using a standardized extraction sheet, which was collaboratively developed and pilot-tested by A.A.L. and A.J.A. Extracted data included study design, country, number of centers, total participants, control group characteristics, inclusion criteria, primary outcomes, and follow-up duration. Baseline characteristics such as patient numbers, age, gender, body mass index (BMI), blood pressure, aneurysm location, grading, medical history, surgical interventions, and the time from symptom onset to enrollment were also recorded.

Efficacy outcomes included discharge mRS score [21], mRS scores [21] at 3 and 6 months, excellent neurological recovery (mRS [21] 0–1), functional independence (mRS [21] 0–2), and independent ambulation (mRS [21] 0–3). Safety outcomes included death at discharge, DCI, hypoglycemia, hydrocephalus, pulmonary infections, and poor neurological recovery (mRS [21] 4–6). Data extraction was performed independently by A.A.L. and A.J.A., and any discrepancies were resolved through discussion to ensure accuracy.

### 2.6. Risk-of-Bias Assessment and Certainty of Evidence

The quality of the included randomized controlled trials (RCTs) was evaluated using the Cochrane Risk of Bias 2 (RoB 2) tool [25]. The assessment covered key domains such as selection, performance, detection, attrition, and reporting biases. Two reviewers, M.M.A and A.J.A., conducted the evaluation independently, and disagreements were resolved through discussion or, if necessary, by a third reviewer, R.B.A.

### 2.7. Statistical Analysis

The meta-analysis employed both common-effect and random-effects models. The Mantel–Haenszel method was used for the common-effect model, while the inverse variance method was applied for the random-effects model. The restricted maximum-likelihood estimator (REML) was used to estimate tau-squared (τ^2^) for assessing heterogeneity. All statistical analyses were conducted using R version 4.3.2 [26].

## 3. Results

### 3.1. Search Results and Study Selection

A total of 107 records were identified through our database search. After removing 30 duplicates, 77 records remained for abstract screening. Following the screening of abstracts and titles, 63 records were excluded, leaving 14 studies for full-text review. Of these, 11 studies were excluded, resulting in 3 studies being included in our review. The search and selection process is illustrated in Figure 2 and Figure 3.

### 3.2. Characteristics of Included Studies

The characteristics of the three included studies are summarized in Table 2. da Costa and colleagues (2022) [18] (Brazil, 78 participants) evaluated the 6-month mRS score [21] distribution following clipping/coiling within 96 h in patients aged 18–70. This study found that 5 mg glibenclamide orally daily for 21 days did not improve 6-month functional outcomes, mortality, or delayed cerebral ischemia rates compared to placebo [21]. Feng and colleagues (2024) [27] (China, 56 participants) assessed the proportion of patients achieving a SEBES [22] of 0–2 at 10 days post-medication in those aged ≥18 with Hunt–Hess grade of subarachnoid hemorrhage [28] ≥2 undergoing surgery within 72 h. The authors reported that high-dose glibenclamide (5 mg orally three times daily for 10 days) significantly reduced cerebral edema at 10 days and improved long-term functional outcomes, although it was associated with increased hypoglycemia [27]. Lin and colleagues (2024) [29] (China, 111 participants) compared serum neuron-specific enolase (NSE) and S100B levels with or without glibenclamide in patients aged 18–74 with aSAH diagnosed within 48 h. Their findings indicated no significant differences in biomarker levels or functional outcomes at 90 days between the glibenclamide (1.25 mg orally three times daily for 7 days) and control groups [29]. Follow-up durations ranged from 3 to 6 months. The characteristics of the study participants in each included study are presented in Table 3.

### 3.3. Risk of Bias and Quality of Evidence

Two RCTs [18,27] exhibited a low overall risk of bias, indicating a high level of methodological rigor. These reliable findings reinforce the validity of the studies and support evidence-based practice (Figure 4).

### 3.4. Efficacy

Using the random-effects model the pooled mean difference (MD) and confidence interval (CI) of the mRS scores [21] were [MD −0.19 with 95% CI (−2.05, 1.66), not significant (NS)] with high heterogeneity (I2 = 79%, *p* = 0.03) (Figure 5A) at discharge, [MD 0.06 with 95% CI (−0.60, 0.71), NS] without heterogeneity (I2 = 0%, *p* = 0.94) at 3 months (Figure 5B), and [MD 0.4 with 95% CI (−0.67, 087), NS] without heterogeneity (I2 = 0%, *p* = 0.9) at 6 months (Figure 5C). Using the random effects model the pooled risk ratio (RR) for poor neurological recovery (mRS [21] 4–6) for all the included studies was [RR 0.91, (0.58, 1.43)] with moderate heterogeneity (I2 = 41%, *p* = 0.19) (Figure 5D).

The pooled estimate of (A) excellent neurological recovery (mRS [21] 0–1) was [RR 1.06 (0.78, 1.46)] with no heterogeneity (I2 = 0%, *p* = 0.59) (Figure 6A), (B) functional independence, [RR 1.05, (0.81, 1.36)] with no heterogeneity (I2 = 0%, *p* = 0.36) (Figure 6B), and (C) independent ambulation, [RR 1.07, (0.77, 1.48)] with moderate heterogeneity (I2 = 57%, *p* = 0.71) (Figure 6C).

### 3.5. Safety

Using the random-effects model the pooled estimate of (A) death was [RR 0.79, (0.42, 1.50)] with no heterogeneity (I2 = 0%, *p* < 0.01) (Figure 7A), (B) delayed cerebral ischemia, [RR 0.58, (0.31, 1.09)] with low heterogeneity (I2 = 31%, *p* = 0.23) (Figure 7B), and (C) hypoglycemia, [RR 3.92, (1.14, 13.49)] without heterogeneity (I2 = 0%, *p* = 0.70) (Figure 7C).

## 4. Discussion

The management of aneurysmal subarachnoid hemorrhage (aSAH) remains a formidable challenge in modern neurology and neurosurgery. Despite advances in aneurysm repair techniques and the introduction of pharmacological agents like nimodipine, outcomes for aSAH patients have not improved significantly over recent decades. Cerebral edema, a frequent and debilitating secondary complication of aSAH, exacerbates ICP and contributes to poor prognoses. This discussion delves into the potential role of glibenclamide as a therapeutic agent in managing aSAH, debates its strengths and limitations, and examines whether its preclinical promise can translate into clinical benefit.

### 4.1. Unmet Needs in aSAH Management

The mortality rate of nearly 44% and the high prevalence of long-term cognitive and neurological dysfunction among survivors reflect the urgent need for novel therapeutic strategies. Current treatment modalities focus primarily on addressing vasospasm and DCI, yet cerebral edema, an equally critical complication, remains inadequately managed. Decompressive craniectomy, a mechanical intervention to reduce ICP, has shown limited benefits in terms of long-term neurological outcomes and often carries significant risks, including infections and brain herniation [7,8]. Nimodipine, while modestly effective in preventing ischemic damage, offers limited efficacy in reducing cerebral edema directly [9].

An unmet need in aSAH management is therefore a lack of effective interventions for the molecular mechanisms underlying cerebral edema. The SUR1-TRPM4 channel has emerged as a promising target in this regard, as its upregulation following ischemic and hemorrhagic insults directly contributes to cytotoxic and vasogenic edema by promoting sodium and water influx into brain tissues [14,15,16]. Glibenclamide, a SUR1-TRPM4 channel blocker, has shown considerable neuroprotective effects in preclinical models [12,13,14,15,16]. However, translating these findings into clinical practice presents significant challenges.

### 4.2. Limited Efficacy of Oral Glibenclamide

#### 4.2.1. Functional Outcomes

Preclinical studies have consistently demonstrated that glibenclamide reduces cerebral edema, preserves endothelial integrity, and mitigates secondary neuronal damage [12,13,14,15,16,17]. However, the clinical evidence from this meta-analysis paints a more complex picture. The pooled results for functional outcomes, as measured using the modified Rankin Scale (mRS) [21], showed no statistically significant improvements in patients treated with oral glibenclamide compared to placebo.

At discharge, the pooled mean difference in mRS [21] was [MD −0.19, (−2.05, 1.66)] with high heterogeneity, reflecting variability in study designs and patient populations.At 3 months, the mRS [21] pooled mean difference was [MD 0.06, (−0.60,0.71)] with no heterogeneity, indicating consistent findings across studies but no significant benefit.

These findings raise the question: why does oral glibenclamide fail to achieve the expected improvements in functional outcomes? One potential explanation lies in the timing of intervention. The therapeutic window for preventing secondary injury and edema formation may be narrower than currently assumed. While preclinical studies suggest a 3- to 10-day window, in clinical practice, delays in diagnosis, imaging, and initiation of therapy may reduce efficacy. Another potential explanation is the limited efficacy of oral glibenclamide due to the problematic pharmacokinetics resulting from glibenclamide taken by mouth. The three articles that met our inclusion criteria all administered glibenclamide by mouth (5 mg orally daily for 21 days [18], 5 mg orally three times daily for 10 days [27], 1.25 mg orally three times daily for 7 days [29]). These findings have been confirmed by another systematic review and meta-analysis [31] that included a study of oral glibenclamide (5 mg daily for 21 days) [32] comparable to those included in the current article. Both the study [32] and the meta-analysis [31] were published after the conclusion of this meta-analysis so they are not included in the current article. By contrast, the continuous intravenous infusion of glibenclamide to patients with large hemispheric ischemic infarctions provided a constant therapeutic dose of the agent without spikes in serum drug levels leading to insulin release and resultant hypoglycemia [33]. The limited efficacy of the studies included in this article may have resulted from the subtherapeutic doses attained by the oral administration of glibenclamide in the studies that met our inclusion criteria [18,27,29].

#### 4.2.2. Neurological Recovery and Independence

The analysis of poor neurological recovery (mRS [21] 4–6) and excellent neurological recovery (mRS [21] 0–1) further underscores the limited impact of oral glibenclamide. The pooled risk ratios of [RR 0.91, (0.58, 1.43)] and [RR 1.06, (0.78, 1.46]) for treatment and placebo groups, respectively, indicate no significant difference between the groups. Similarly, measures of functional independence and ambulation revealed no meaningful benefits. These findings suggest that while oral glibenclamide may mitigate cerebral edema, this alone may not suffice to improve complex neurological and functional outcomes. By contrast, intravenous glibenclamide demonstrated a trend for benefit after treatment for 90 days [33].

### 4.3. Safety Profile

A key concern with glibenclamide is its association with hypoglycemia, a known side effect of sulfonylureas. This systematic review identified a significantly higher risk of hypoglycemia in the oral glibenclamide group [RR 3.92, (1.14, 13.49)]. Hypoglycemia can exacerbate neurological injury in critically ill patients, particularly those with aSAH, by compromising cerebral glucose metabolism. This risk necessitates careful patient monitoring and consideration of alternative dosing regimens or formulations, such as local delivery methods, to minimize systemic side effects. The peaks in serum glibenclamide resulting from oral administration stimulate insulin release, resulting in hypoglycemia. This unfavorable effect of the pharmacokinetics of oral glibenclamide can be ameliorated by the continuous infusion of intravenous glibenclamide to produce a steady-state serum level of glibenclamide without insulin release [33].

On a positive note, oral glibenclamide did not significantly increase the risk of mortality [RR 0.79, (0.42, 1.50)] or delayed cerebral ischemia [RR 0.58, (0.31, 1.09)], suggesting that it is generally well tolerated when administered under controlled conditions. The lack of significant harm strengthens the case for further exploration, albeit with strategies to mitigate hypoglycemia.

### 4.4. Mechanism of Action

One of the major debates surrounding glibenclamide is whether its mechanism of action is sufficient to address the multifaceted pathophysiology of aSAH. While the SUR1-TRPM4 channel is a critical mediator of edema, aSAH-induced brain injury involves complex and interconnected pathways, including inflammation, oxidative stress, and apoptosis [17,18]. Critics argue that targeting a single channel may offer only limited benefits, particularly in a condition as multifactorial as aSAH.

Supporters of glibenclamide counter that its effects extend beyond edema reduction. By preserving endothelial integrity and reducing inflammation, glibenclamide may indirectly mitigate other injury pathways [12,18]. This broader cytoprotective effect could explain its success in preclinical models but raises questions about its translation to human subjects, where variability in comorbidities and injury severity may dilute its impact.

### 4.5. Translation of Preclinical Findings to Clinical Practice

Another contentious issue is the discrepancy between preclinical and clinical results. Animal models of aSAH often involve controlled and homogenous injuries, whereas human aSAH presents with significant variability in aneurysm size, location, and rupture severity. This heterogeneity may partly explain why glibenclamide’s benefits in preclinical studies have not translated into clinical success [18]. Moreover, the preclinical studies often used higher doses of glibenclamide and initiated treatment earlier, conditions that are challenging to replicate in clinical practice. Future clinical trials of glibenclamide may be improved to provide a constant therapeutic dose by providing a continuous intravenous infusion or frequent oral dosages. Preclinical investigations may help to fine-tune the safety and efficacy of these stratagies.

### 4.6. Timing of Intervention

Timing is a critical factor in the efficacy of any pharmacological intervention. Some researchers argue that the 3- to 10-day window proposed for glibenclamide intervention may already be too late to prevent irreversible neuronal damage. Studies indicate that significant brain injury mechanisms, including cytotoxic edema and early inflammatory responses, occur within the first 48 h following aSAH onset, suggesting a need for even earlier intervention [12,17,18]. Conversely, prolonged treatment beyond this window may mitigate delayed complications such as vasospasm or DCI, as evidenced by improved outcomes in animal models when glibenclamide was administered over extended periods [17]. Determining the optimal timing and duration of therapy is critical to maximizing glibenclamide’s therapeutic potential.

### 4.7. Limitations

This meta-analysis synthesizes a broad spectrum of clinical evidence, offering valuable insights into the safety and efficacy of oral glibenclamide. It addresses key outcomes, including functional independence, neurological recovery, and mortality, thereby providing a holistic view of the therapy’s impact. By leveraging pooled data, the review reduces individual study biases and enhances statistical power [19,20].

However, several limitations to the current meta-analysis must be acknowledged. The included studies exhibited variability in their designs, dosing regimens, and outcome measures, contributing to heterogeneity in some analyses. The relatively small number of randomized controlled trials (RCTs) and limited sample sizes in individual studies restrict the robustness of conclusions [7,18,27,29,31,32]. Furthermore, the lack of long-term follow-up data prevents a thorough assessment of sustained benefits or risks associated with glibenclamide use. Additionally, the absence of biomarker-guided patient selection may obscure potential benefits in specific subgroups [12,17].

The slow progress in developing effective therapies for aSAH, including glibenclamide, can be attributed to several factors:Complex Pathophysiology: The multifaceted nature of aSAH-induced brain injury, involving edema, inflammation, vasospasm, and apoptosis, makes it challenging to develop targeted therapies. Therapies like glibenclamide, which focus on a single pathway, may fail to address the broader spectrum of injury mechanism [12,16].Heterogeneity of patients: Variability in patient demographics, aneurysm location and size, and comorbidities introduces confounding factors in clinical trials. Studies show that responses to treatments can differ significantly based on these variables, making it difficult to achieve uniform outcomes [7,18,25,27,31,32].Translation Challenges: Differences in dosing regimens, injury models, and treatment timelines between preclinical and clinical studies often result in inconsistent outcomes. For example, glibenclamide doses used in animal models are frequently higher and initiated earlier, compared to clinical settings [12,17,18,25,27,31,32].Limited Resources: The high costs and complexity of conducting large-scale, multicenter trials for aSAH therapies further hinder progress. This is compounded by the rarity of aSAH compared to other stroke subtypes, making patient recruitment a challenge [3,4,7].

These factors emphasize the need for innovative trial designs, biomarker-guided therapies, and a better understanding of aSAH pathophysiology to overcome these barriers and accelerate progress.

## 5. Conclusions

Glibenclamide holds promise as a therapeutic agent in aSAH, offering a novel mechanism of action targeting cerebral edema. While its safety profile is reassuring, the lack of significant improvements in functional and neurological outcomes achieved by the oral administration of glibenclamide raises critical questions about its role as a standalone therapy. Addressing these challenges through innovative trial designs and combination therapies could unlock its potential and pave the way for better outcomes in aSAH patients. The journey to effective therapies for aSAH is far from over, but glibenclamide represents a step in the right direction. The current systematic review and meta-analysis provides evidence for the need for multiple RCTs with uniform protocols, including ratings at specified time periods across a spectrum of international centers, involving participants of a wide range of ethnicities. In this meta-analysis, oral glibenclamide did not show significant benefits in improving functional outcomes, promoting neurological recovery, or reducing mortality. However, its use was associated with a notably higher risk of hypoglycemia, highlighting the importance of cautious monitoring. Future systematic reviews and meta-analyses with many well-designed and well-executed investigations utilizing optimal protocols, including continuous intravenous infusions to attain therapeutic doses without stimulating insulin release [33] can resolve some of the current uncertainties about the safety and efficacy of glibenclamide for aSAH.

## Figures and Tables

**Figure 1 brainsci-15-00677-f001:**
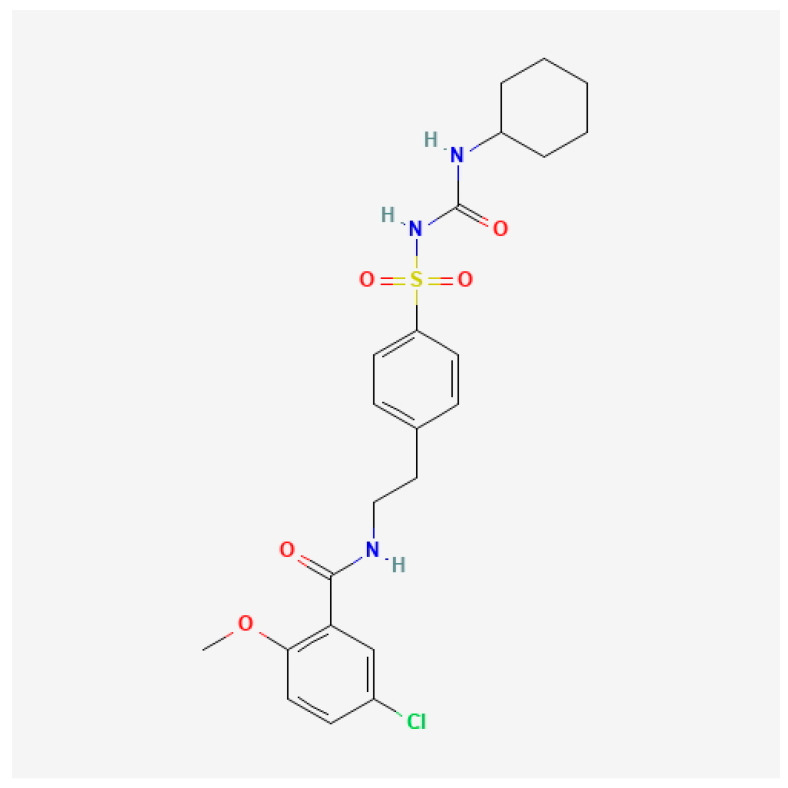
Glibenclamide chemical structure depiction: a 2D structure image of CID 129848290, glibenclamide hydrochloride [10].

**Figure 2 brainsci-15-00677-f002:**
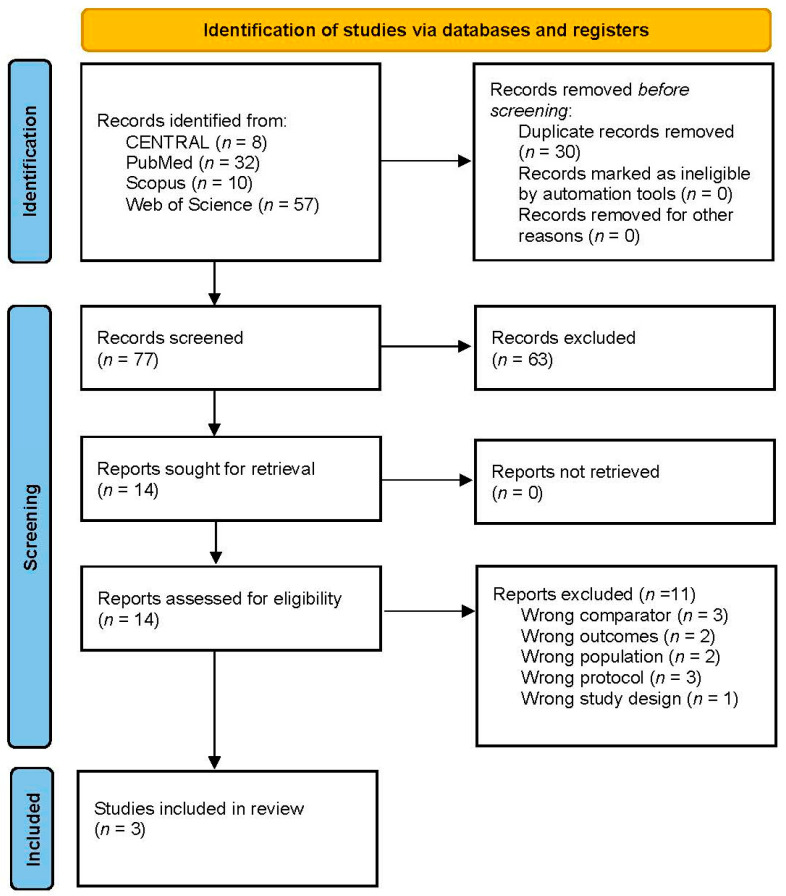
Preferred Reporting Items for Systematic Reviews and Meta-Analyses (PRISMA) [19] flowchart of the screening process for glibenclamide for cerebral edema after aneurysmal subarachnoid hemorrhage. Source: [15] Page MJ, et al. BMJ 2021;372:n71. doi: 10.1136/bmj.n71. This work is licensed under CC BY 4.0. Available online: https://creativecommons.org/licenses/by/4.0/ (accessed on 23 June 2025).

**Figure 3 brainsci-15-00677-f003:**
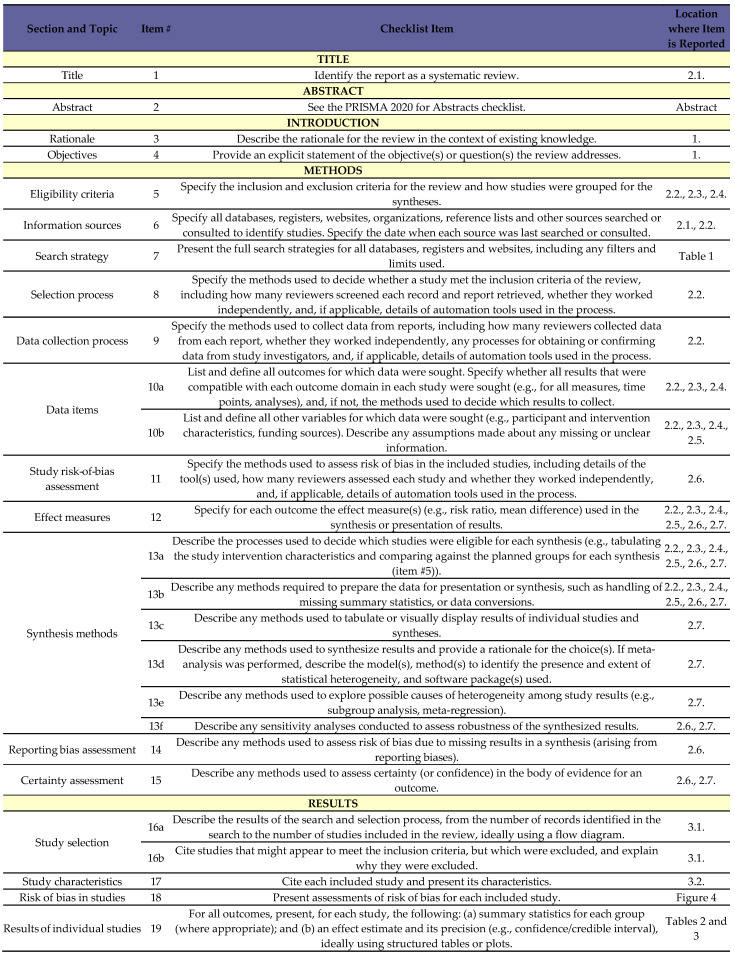
Preferred Reporting Items for Systematic Reviews and Meta-Analyses (PRISMA) [19] checklist of the screening process for glibenclamide for cerebral edema after aneurysmal subarachnoid hemorrhage. *Source*: Page MJ, McKenzie JE, Bossuyt PM, et al. The PRISMA 2020 statement: an updated guideline for reporting systematic reviews. BMJ 2021;372:n71. doi:10.1136/bmj.n71 [19].

**Figure 4 brainsci-15-00677-f004:**
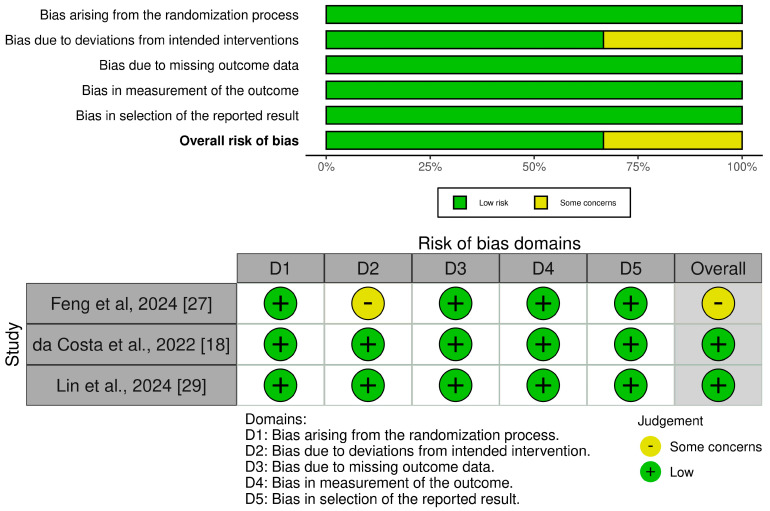
Risk-of-bias assessment of studies included in a systematic review and meta-analysis of glibenclamide for cerebral edema after aneurysmal subarachnoid hemorrhage [18,27,29].

**Figure 5 brainsci-15-00677-f005:**
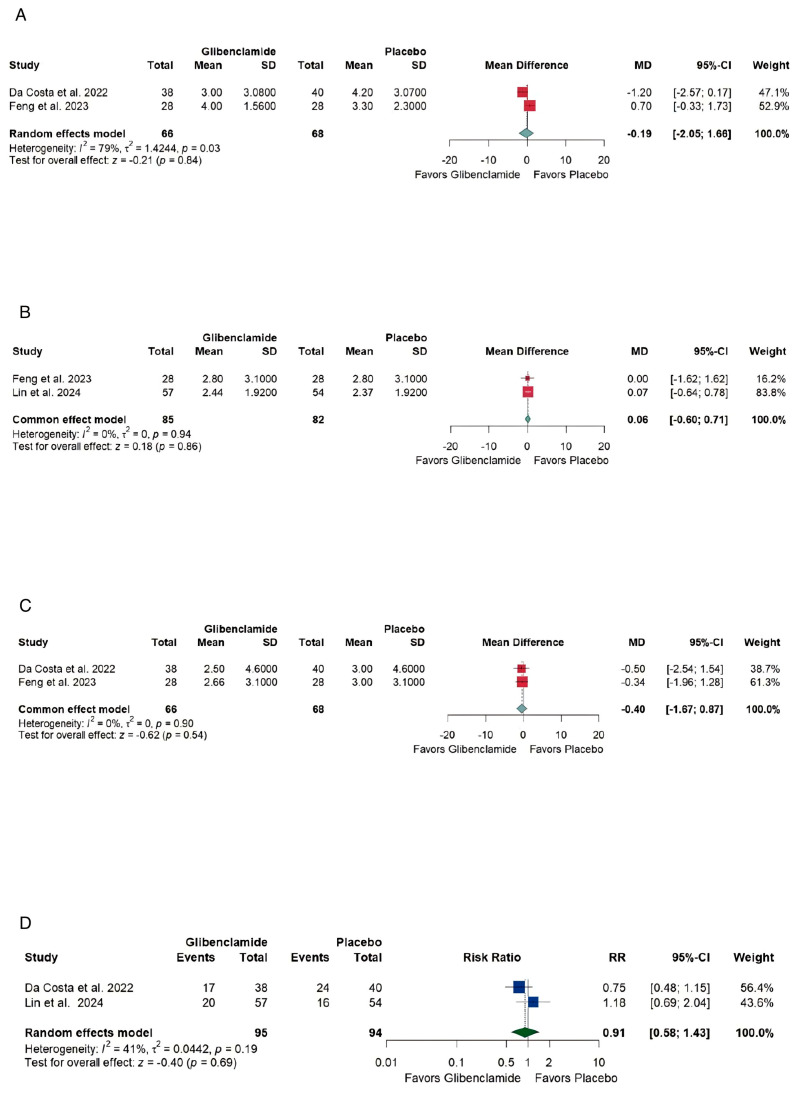
Forest plots of outcomes of a systematic review and meta-analysis of glibenclamide for cerebral edema after aneurysmal subarachnoid hemorrhage. (**A**) Discharge mRS [18,21,27], (**B**) 3 months mRS [18,21,29], (**C**) 6 months mRS [18,21,27], (**D**) Poor neurological recovery (mRS [18,21,29], 4–6).

**Figure 6 brainsci-15-00677-f006:**
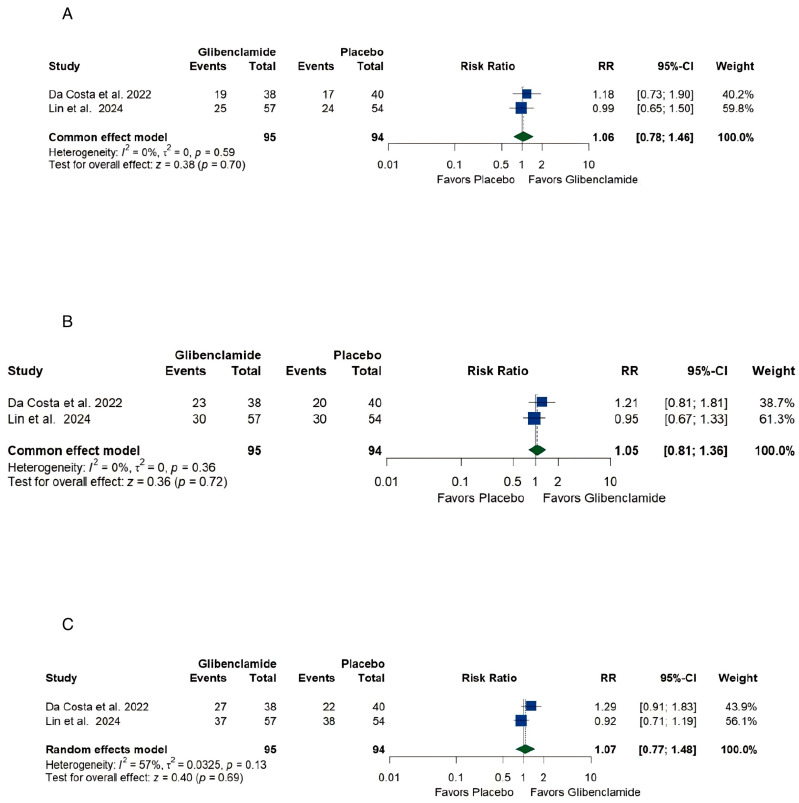
Forest plots of outcomes of a systematic review and meta-analysis of glibenclamide for cerebral edema after aneurysmal subarachnoid hemorrhage. (**A**) Excellent neurological recovery (mRS [21] 0–1) [18,29], (**B**) Functional independence [18,29], (**C**) Independent ambulation [18,29].

**Figure 7 brainsci-15-00677-f007:**
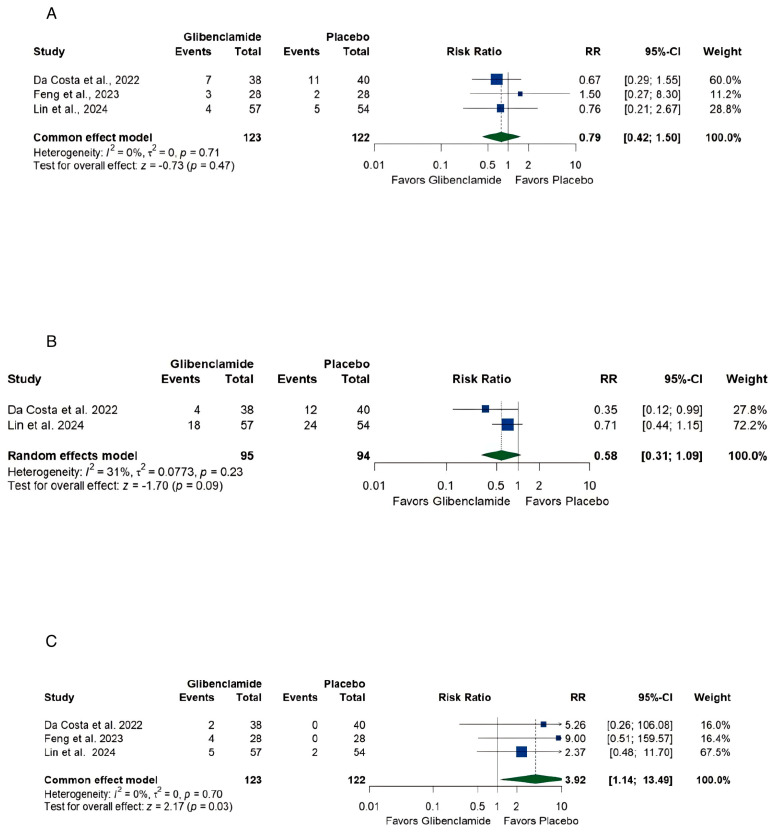
Forest plots of outcomes of a systematic review and meta-analysis of glibenclamide for cerebral edema after aneurysmal subarachnoid hemorrhage. (**A**) Death [18,27,29], (**B**) Delayed cerebral ischemia [18,29], (**C**) Hypoglycemia [18,27,29].

**Table 1 brainsci-15-00677-t001:** Detailed database search strategy for a systematic review and meta-analysis of glibenclamide for cerebral edema after aneurysmal subarachnoid hemorrhage.

Database	Search Terms	Search Field	Search Results
Pubmed	(Glibenclamide OR Glyburide OR “KATP channel blocker” OR Sulphonylurea) AND (“Subarachnoid Hemorrhage” OR “Subarachnoid Haemorrhage” OR SAH OR “Aneurysmal Subarachnoid Hemorrhage” OR “Aneurysmal Subarachnoid Haemorrhage”)	All Field	32
Cochrane	(Glibenclamide OR Glyburide OR “KATP channel blocker” OR Sulphonylurea) AND (“Subarachnoid Hemorrhage” OR “Subarachnoid Haemorrhage” OR SAH OR “Aneurysmal Subarachnoid Hemorrhage” OR “Aneurysmal Subarachnoid Haemorrhage”)	All Text	8
WOS	(Glibenclamide OR Glyburide OR “KATP channel blocker” OR Sulphonylurea) AND (“Subarachnoid Hemorrhage” OR “Subarachnoid Haemorrhage” OR SAH OR “Aneurysmal Subarachnoid Hemorrhage” OR “Aneurysmal Subarachnoid Haemorrhage”)	All Fields	57
SCOPUS	“glibenclamide OR glyburide OR “KATP channel blocker” AND “Subarachnoid Hemorrhage” OR “Subarachnoid Haemorrhage” OR SAH OR “Aneurysmal Subarachnoid Hemorrhage” OR “Aneurysmal Subarachnoid Haemorrhage”	Title, Abstract, Keywords	60

**Table 2 brainsci-15-00677-t002:** Characteristics of studies included in a systematic review and meta-analysis of glibenclamide for cerebral edema after aneurysmal subarachnoid hemorrhage.

First Author/ Year	Study Design	Country	Centers	Total Participants	Control	Main Inclusion Criteria	Primary Outcome	Follow-up Duration
da Costa et al., 2022 [18]	Randomized, double-blind and prospective clinical trial.	1 country (Brazil)	Single center (Hospital das Clínicas da Faculdade de Medicina da Universidade de São Paulo, at São Paulo)	78	Placebo	Inclusion: SAH confirmed radiologically, aneurysmal origin verified, age 18–70, and treatment (clipping/coiling) within 96 h	6-month mRS score distribution	6 months
Feng et al., 2024 [27]	Randomized, double-blind, placebo controlled clinical trial.	1 country (China)	Single center (Xuanwu Hospital Capital Medical University, Beijing, China)	56	Placebo	Inclusion: Radiological aSAH, age ≥ 18, surgery within 72 h, Hunt–Hess grade ≥ 2	Proportion of patients with SAH Early Brain Edema Score 0–2 at 10 days post-medication.	3 and 6 months
Lin et al., 2024 [29]	Randomized, controlled, open-label, blinded- endpoint clinical trial.	1 country (China)	Multicenter (Beijing Tiantan Hospital)	111	Neither glibenclamide tablets nor placebo	Inclusion: Radiological aSAH within 48 h, age 18–74 (older age due to lower hypoglycemia tolerance)	Difference in serum NSE and S100B levels with or without glibenclamide	3 months

**Table 3 brainsci-15-00677-t003:** Characteristics of participants in studies included in a systematic review and meta-analysis of glibenclamide for cerebral edema after aneurysmal subarachnoid hemorrhage.

Data	Number of Patients in Each Group		Age (Years) Mean (SD)		Male *n* (%)		BMI (kg/m²) Mean (SD)		Related Grading Median (IQR)										Medical History *n* (%)				Surgery *n* (%)				Time from Onset to Enrolment (h), Median (IQR)	
									Hunt-Hess grade [28]		WFNS grade [30]		mFS [23]		SEBES [22]		mRS [21]		Hypertension		Diabetes		Coiling		Clipping			
	G	P	G	P	G	P	G	P	G	P	G	P	G	P	G	P	G	P	G	P	G	P	G	P	G	P	G	P
da Costa et al., 2022 [18]	38	40	53.6 (11.6)	52.7 (11.3)	6 (15.8)	13 (32.5)	NA	NA	3 (2–4)	3 (2–4)	3 (1–4)	2.5 (1–4)	3 (3–4)	3 (3–4)	NA	NA	NA	NA	NA	NA	NA	NA	18 (47.4)	10 (25.0)	20 (52.6)	30 (75.0)	60 (24–96)	72 (48–96)
Feng et al., 2024 [27]	28	28	61.8 (11.6)	59.1 (12.6)	12 (42.9)	17 (60.7)	24.3 (3.6)	25.6 (4.7)	3 (3–4)	3 (3–4)	5 (4–5)	4 (4–5)	4 (3–4)	4 (3–4)	4 (3–4)	4 (2–4)	5 (4.3–5)	4 (2–5)	23 (82.1)	18 (64.3)	6 (21.4)	2 (7.1)	NA	NA	NA	NA	NA	NA
Lin et al., 2024 [29]	57	54	57 (11.4)	55.3 (11.04)	28 (49)	25 (46)	24.5 (2.5)	24.3 (3.02)	3 (3–4)	3 (2–4)	4 (2–4)	4 (2–4)	2 (2–3)	2 (2–3)	2 (2–3)	2 (2–3)	NA	NA	35 (61)	27 (50)	6 (11)	4 (7)	3 (5)	2 (4)	54 (95)	52 (96)	24 (24, 28)	26 (20, 40.75)

BMI—Body mass index; G—Glibenclamide; IQR—Interquartile range; mFS—Modified Fisher Scale [23]; mRS—Modified Rankin Scale [21]; *n*—Number; NA—Not available; P—Placebo; SD—Standard deviation; SEBES—Subarachnoid Hemorrhage Early Brain Edema Score [22]; WFNS—World Federation of Neurological Surgeons Committee on a Universal Subarachnoid Hemorrhage Grading Scale [30].

## Data Availability

All data are included in the text.

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
