# Peer review of "The Safety and Efficacy of Glibenclamide in Managing Cerebral Edema After Aneurysmal Subarachnoid Hemorrhage: A Systematic Review and Meta-Analysis"

_brainsci, 2025, doi:10.3390/brainsci15070677_

Round 1
Reviewer 1 Report
Comments and Suggestions for Authors
This manuscript analyses 3 trials conducted with oral glibenclamide in patients with aSAH. Overall, the review is well written and very credible.
I have a single but important critique that needs to be addressed – the authors omitted the critical issue of glibenclamide dosing and route of administration in the papers reviewed. These factors bear critically on pharmacokinetics and thus have important implications for efficacy.
To date, the most promising data with glibenclamide in humans is in patients with ischemic stroke (large hemispheric infarction) treated with intravenous glibenclamide where swelling (midline shift) and clinical outcomes are improved by drug (e.g., PMID: 29789393). A constant intravenous infusion is critical to maintain steady-state therapeutic levels and to avoid spikes in serum concentration of drug that cause release of insulin that risks hypoglycemia.
The 3 papers reviewed utilized po glibenclamide (DaCosta: 5 mg tablet po q day x 21 days; Feng: 5 mg tablet po tid x 10 days; Lin: 1.25 mg tablet po tid x 7 days), which has very different, and undesirable, pharmacokinetics compared to IV glibenclamide. Pharmacokinetics are critically important with this drug.
My recommendation is that these issues should be carefully addressed in the Discussion. You should point that it is likely that subtherapeutic dosing and an inadequate route of administration may have contributed strongly to the failure of the 3 trials, and that declaring the drug a failure requires clinical trials in which adequate consideration is given to pharmacokinetic issues .
Reviewer 2 Report
Comments and Suggestions for Authors
This manuscript presents a well-structured investigation on a clinically relevant topic, with a high-quality analysis of the available data. However, I have a few comments that could help improve the clarity and overall quality of the manuscript.
1. In Table 3, I recommend including the appropriate measurement units in the column headers to improve clarity. For example: "BMI (kg/m²) Mean (SD)"
2. In the section describing the characteristics of the included studies, it would be helpful if the authors added one sentence summarizing the key finding of each study after its description.
3. Some figures (e.g., Figures 4, 5 and 6) could be presented in higher resolution to improve the readability of numbers and labels.
4. I suggest briefly summarizing the key quantitative findings (e.g., increased risk of hypoglycemia) to reinforce the main takeaways for the reader.
Round 2
Reviewer 1 Report
Comments and Suggestions for Authors
the authors have adequately addressed my concerns